# SURF: Semi-supervised Reward Learning with Data Augmentation for Feedback-efficient Preference-based Reinforcement Learning

**Jongjin Park**[1] **Younggyo Seo**[1] **Jinwoo Shin**[1] **Honglak Lee**[2,4] **Pieter Abbeel**[3] **Kimin Lee**[3]
[1]KAIST [2]University of Michigan [3]UC Berkeley [4]LG AI Research

## Abstract

Preference-based reinforcement learning (RL) has shown potential for teaching agents to perform the target tasks without a costly, pre-defined reward function by learning the reward with a supervisor's preference between the two agent behaviors. However, preference-based learning often requires a large amount of human feedback, making it difficult to apply this approach to various applications. This data-efficiency problem, on the other hand, has been typically addressed by using unlabeled samples or data augmentation techniques in the context of supervised learning. Motivated by the recent success of these approaches, we present SURF, a semi-supervised reward learning framework that utilizes a large amount of unlabeled samples with data augmentation. In order to leverage unlabeled samples for reward learning, we infer pseudo-labels of the unlabeled samples based on the confidence of the preference predictor. To further improve the label-efficiency of reward learning, we introduce a new data augmentation that temporally crops consecutive subsequences from the original behaviors. Our experiments demonstrate that our approach significantly improves the feedback-efficiency of the state-of-the-art preference-based method on a variety of locomotion and robotic manipulation tasks.

## 1 Introduction

Reward function plays a crucial role in reinforcement learning (RL) to convey complex objectives to agents. For various applications, where we can design an informative reward function, RL with deep neural networks has been used to solve a variety of sequential decision-making problems, including board games (Silver et al., 2017; 2018), video games (Mnih et al., 2015; Berner et al., 2019; Vinyals et al., 2019), autonomous control (Schulman et al., 2015; Bellemare et al., 2020), and robotic manipulation (Kober & Peters, 2011; Kober et al., 2013; Kalashnikov et al., 2018; Andrychowicz et al., 2020). However, there are several issues in reward engineering. First, designing a suitable reward function requires more human effort as the tasks become more complex. For example, defining a reward function for book summarization (Wu et al., 2021) is non-trivial because it is hard to quantify the quality of summarization in a scale value. Also, it has been observed that RL agents could achieve high returns by discovering undesirable shortcuts if the hand-engineered reward does fully specify the desired task (Amodei et al., 2016; Hadfield-Menell et al., 2017; Lee et al., 2021a). Furthermore, there are various domains, where a single ground-truth function does not exist, and thus personalization is required by modeling different reward functions based on the user's preference.

Preference-based RL (Akrour et al., 2011; Christiano et al., 2017; Ibarz et al., 2018; Lee et al., 2021a) provides an attractive alternative to avoid reward engineering. Instead of assuming a hand-engineered reward function, a (human) teacher provides preferences between the two agent behaviors, and an agent learns how to show the desired behavior by learning a reward function, which is consistent with the teacher's preferences. Recent progress of preference-based RL has shown that the teacher can guide the agent to perform novel behaviors (Christiano et al., 2017; Stiennon et al., 2020; Wu et al., 2021), and mitigate the effects of reward exploitation (Lee et al., 2021a). However, existing preference-based approaches often suffer from expensive labeling costs, and this makes it hard to apply preference-based RL to various applications.

Meanwhile, recent state-of-the-art system in computer vision, the label-efficiency problem has been successfully addressed through semi-supervised learning (SSL) approaches (Berthelot et al., 2019; 2020; Sohn et al., 2020; Chen et al., 2020b). By leveraging unlabeled dataset, SSL methods have improved the performance with low cost. Data augmentation also plays a significant role in improving the performance of supervised learning methods (Cubuk et al., 2018; 2019). By using multiple augmented views of the same data as input, the performance has been improved by learning augmentation-invariant representations.

Inspired by the impact of semi-supervised learning and data augmentation, we present SURF: a **S**emi-s**U**pervised **R**eward learning with data augmentation for **F**eedback-efficient preference-based RL. To be specific, SURF consists of the following key ingredients:

(a) Pseudo-labeling (Lee, 2013; Sohn et al., 2020): We leverage unlabeled data by utilizing the artificial labels generated by learned preference predictor, which makes the reward function produce a confident prediction (see Figure 1a). We remark that such a SSL approach is particularly attractive in our setup as an unlimited number of unlabeled data can be obtained with no additional cost, i.e., from past experiences stored in the buffer.

(b) Temporal cropping augmentation: We generate slightly shifted or resized behaviors, which are expected to have the same preferences from a teacher, and utilize them for reward learning (see Figure 1b). Our data augmentation technique enhances the feedback-efficiency by enforcing consistencies (Xie et al., 2019; Berthelot et al., 2020; Sohn et al., 2020) to the reward function.

We remark that SURF is not a naïve application of these two techniques, but a novel combination of semi-supervised learning and the proposed data augmentation, which has not been considered or evaluated in the context of the preference-based RL.

Our experiments demonstrate that SURF significantly improves the preference-based RL method (Lee et al., 2021a) on complex locomotion and robotic manipulation tasks from DeepMind Control Suite (Tassa et al., 2018; 2020) and Meta-world (Yu et al., 2020), in terms of feedback-efficiency. In particular, our framework could make RL agents achieve $\sim$100% of success rate on complex robotic manipulation task using only a few hundred preference queries, while its baseline method only achieves $\sim$50% of the success rate under the same condition (see Figure 2). Furthermore, we show that SURF can improve the performance of preference-based RL algorithms when we operate on high-dimensional and partially-observable inputs.

## 2 RELATED WORK

**Preference-based RL**. In the preference-based RL framework, a (human) supervisor provides preferences between the two agent behaviors and the agent uses this feedback to perform the task (Christiano et al., 2017; Ibarz et al., 2018; Leike et al., 2018; Stiennon et al., 2020; Wu et al., 2021; Lee et al., 2021a;b). Since this approach is only feasible if the feedback is practical for a human to provide, several strategies have been studied in the literature. Ibarz et al. (2018) initialized the agent's policy with imitation learning from the expert demonstrations, while Lee et al. (2021a) utilized unsupervised pre-training for policy initialization. Several sampling schemes (Sadigh et al., 2017; Biyik & Sadigh, 2018; Biyik et al., 2020) to select informative queries also have been adopted for improving the feedback-efficiency. Our approach differs in that we utilize unlabeled samples for reward learning, and also provide a novel data augmentation technique for the agent behaviors.

**Data augmentation for RL**. In the context of RL, data augmentation has been widely investigated for improving data-efficiency (Srinivas et al., 2020; Yarats et al., 2021), or RL generalization (Cobbe et al., 2019; Lee et al., 2019). For example, RAD (Laskin et al., 2020) demonstrated that data augmentation, such as random crop, can improve both data-efficiency and generalization of RL algorithms. While these methods are known to be beneficial to learn policy in the standard RL setup, they have not been tested for learning *rewards*. To the best of our knowledge, we present the first data augmentation method specially designed for learning reward function.

**Semi-supervised learning**. The goal of semi-supervised learning (SSL) is to leveraging unlabeled samples to improve a model's performance when the amount of labeled samples are limited. In an attempt to leverage the information in the unlabeled dataset, a number of techniques have been proposed, e.g., entropy minimization (Grandvalet & Bengio, 2004; Lee, 2013) and consistency regularization (Sajjadi et al., 2016; Miyato et al., 2018; Xie et al., 2019; Sohn et al., 2020). Recently,

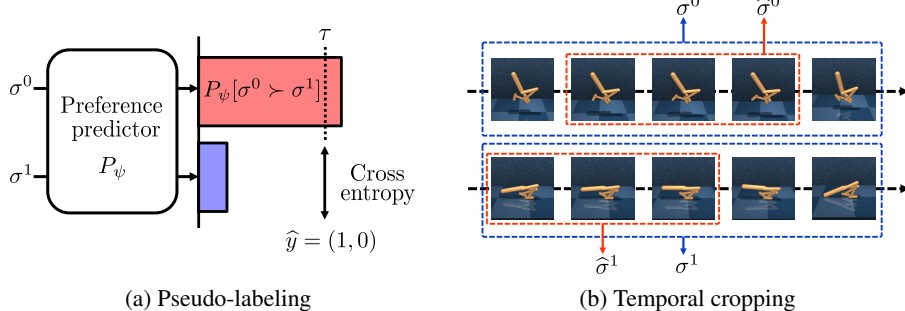

(a) Pseudo-labeling                          (b) Temporal cropping

Figure 1: Overview of SURF. (a) We leverage unlabeled experiences by generating pseudo-labels $\widehat{y}$ from the preference predictor $P_\psi$ in (1). To mitigate the negative effects from this semi-supervised learning, we only utilize pseudo-labels when the confidence of the predictor is higher than threshold $\tau$. (b) Given two segments $(\sigma^0, \sigma^1)$, we generate augmented segments $(\widehat{\sigma}^0, \widehat{\sigma}^1)$ by cropping the subsequence from each segment.

the combination of these two approaches have shown state-of-the-art performance in benchmarks, e.g., MixMatch (Berthelot et al., 2019), and ReMixMatch (Berthelot et al., 2020), when used with advanced data augmentation techniques (Zhang et al., 2018; Cubuk et al., 2019). Specifically, Fix-Match (Sohn et al., 2020) revisits pseudo-labeling technique and demonstrates that joint usage of pseudo-labels and consistency regularization achieves remarkable performance due to its simplicity.

## 3 PRELIMINARIES

Reinforcement learning (RL) is a framework where an agent interacts with an environment in discrete time (Sutton & Barto, 2018). At each timestep $t$, the agent receives a state $\mathbf{s}_t$ from the environment and chooses an action $\mathbf{a}_t$ based on its policy $\pi(\mathbf{a}_t|\mathbf{s}_t)$. In conventional RL framework, the environment gives a reward $r(\mathbf{s}_t, \mathbf{a}_t)$ and the agent transitions to the next state $\mathbf{s}_{t+1}$. The return $\mathcal{R}_t = \sum_{k=0}^{\infty} \gamma^k r(\mathbf{s}_{t+k}, \mathbf{a}_{t+k})$ is defined as discounted cumulative sum of the reward with discount factor $\gamma \in [0, 1)$. The goal of the agent is to learn a policy that maximizes the expected return.

**Preference-based reinforcement learning**. In this paper, we consider a preference-based RL framework, which does not assume the existence of hand-engineered reward. Instead, a (human) teacher provides preferences between the agent's behaviors and the agent uses this feedback to perform the task (Christiano et al., 2017; Ibarz et al., 2018; Leike et al., 2018; Stiennon et al., 2020; Lee et al., 2021a;b; Wu et al., 2021) by learning a reward function, which is consistent with the observed preferences.

We formulate a reward learning problem as a supervised learning problem (Wilson et al., 2012; Christiano et al., 2017). Formally, a segment $\sigma$ is a sequence of observations and actions $\{(\mathbf{s}_k, \mathbf{a}_k), ..., (\mathbf{s}_{k+H-1}, \mathbf{a}_{k+H-1})\}$. Given a pair of segments $(\sigma^0, \sigma^1)$, a teacher gives a feedback indicating which segment is preferred, i.e., $y \in \{0, 1, 0.5\}$, where 1 indicates $\sigma^1 \succ \sigma^0$, 0 indicates $\sigma^0 \succ \sigma^1$, and 0.5 implies an equally preferable case. Each feedback is stored in a dataset $\mathcal{D}$ as a triple $(\sigma^0, \sigma^1, y)$. Then, we model a preference predictor using the reward function $\widehat{r}_\psi$ following the Bradley-Terry model (Bradley & Terry, 1952):

$$P_\psi[\sigma^1 \succ \sigma^0] = \frac{\exp(\sum_t \widehat{r}_\psi(\mathbf{s}_t^1, \mathbf{a}_t^1))}{\sum_{i \in \{0,1\}} \exp(\sum_t \widehat{r}_\psi(\mathbf{s}_t^i, \mathbf{a}_t^i))}, \tag{1}$$

where $\sigma^i \succ \sigma^j$ denotes the event that segment $i$ is preferable to segment $j$. The underlying assumption of this model is that the teacher's probability of preferring a segment depends exponentially on the accumulated sum of the reward over the segment. The reward model is trained through supervised learning with teacher's preferences. Specifically, given a dataset of preferences $\mathcal{D}$, the reward function is updated by minimizing the binary cross-entropy loss:

$$\mathcal{L}^{\texttt{CE}} = \mathop{\mathbb{E}}_{(\sigma^0, \sigma^1, y) \sim \mathcal{D}} \left[ \mathcal{L}^{\texttt{Reward}} \right] = - \mathop{\mathbb{E}}_{(\sigma^0, \sigma^1, y) \sim \mathcal{D}} \left[ (1 - y) \log P_\psi[\sigma^0 \succ \sigma^1] + y \log P_\psi[\sigma^1 \succ \sigma^0] \right].$$

The reward function $\widehat{r}_\psi$ is usually optimized only using labels from real human, which are expensive to obtain in practice. Instead, we propose a simple yet effective method based on semi-supervised learning and data augmentation to improve the feedback-efficiency of preference-based learning.

---

**Algorithm 1** SURF

---

**Require:** Hyperparameters: unlabeled batch ratio $\mu$, threshold parameter $\tau$, and loss weight $\lambda$
**Require:** Set of collected labeled data $\mathcal{D}_l$, and unlabeled data $\mathcal{D}_u$
1: **for** each gradient step **do**
2:     Sample labeled batch $\{(\sigma_l^0, \sigma_l^1, y)^{(i)}\}_{i=1}^B \sim \mathcal{D}_l$
3:     Sample unlabeled batch $\{(\sigma_u^0, \sigma_u^1)^{(j)}\}_{j=1}^{\mu B} \sim \mathcal{D}_u$
4:     // DATA AUGMENTATION FOR LABELED DATA
5:     **for** $i$ in $1 \ldots B$ **do**
6:         $(\widehat{\sigma}_l^0, \widehat{\sigma}_l^1)^{(i)} \leftarrow \mathrm{TDA}((\sigma_l^0, \sigma_l^1)^{(i)})$ in Algorithm 2
7:     **end for**
8:     // PSEUDO-LABELING AND DATA AUGMENTATION FOR UNLABELED DATA
9:     **for** $j$ in $1 \ldots \mu B$ **do**
10:        Predict pseudo-labels $\widehat{y}((\sigma_u^0, \sigma_u^1)^{(j)})$
11:        $(\widehat{\sigma}_u^0, \widehat{\sigma}_u^1)^{(j)} \leftarrow \mathrm{TDA}((\sigma_u^0, \sigma_u^1)^{(j)})$ in Algorithm 2
12:     **end for**
13:     Optimize $\mathcal{L}^{\mathrm{SSL}}$ (3) with respect to $\psi$
14: **end for**

---

## 4 SURF

In this section, we present SURF: a **S**emi-s**U**pervised **R**eward learning with data augmentation for **F**eedback-efficient preference-based RL, that can be used in conjunction with any existing preference-based RL methods. Our main idea is to leverage a large number of unlabeled samples collected from environments for reward learning, by inferring pseudo-labels. To further increase the effective number of training samples, we propose a new data augmentation that temporally crops the subsequence of the agent behaviors. The full procedure of our unified framework in Algorithm 1 (See Figure 1 for the overview of our method).

### 4.1 SEMI-SUPERVISED REWARD LEARNING

To improve the feedback efficiency, we propose a semi-supervised learning (SSL) method for leveraging unlabeled experiences in the buffer for reward learning. In addition to a *labeled dataset* $\mathcal{D}_l = \{(\sigma_l^0, \sigma_l^1, y)^{(i)}\}_{i=1}^{N_l}$, we utilize an *unlabeled dataset* $\mathcal{D}_u = \{(\sigma_u^0, \sigma_u^1)^{(i)}\}_{i=1}^{N_u}$ to optimize the reward model $r_\psi$.[1] Specifically, we generate the artificial labels $\widehat{y}$ by *pseudo-labeling* (Lee, 2013; Sohn et al., 2020) for the unlabeled dataset $\mathcal{D}_u$. We infer a preference $\widehat{y}$ for an unlabeled segment pair $(\sigma_u^0, \sigma_u^1)$ as a class with higher probability as follows:

$$\widehat{y}(\sigma_u^0, \sigma_u^1) = \begin{cases} 0, & \text{if } P_\psi[\sigma_u^0 \succ \sigma_u^1] > 0.5 \\ 1, & \text{otherwise.} \end{cases} \tag{2}$$

By generating labels from the prediction model, we can obtain free supervision for optimizing our reward model. However, pseudo-labels from low-confidence predictions can be inaccurate, and such noisy feedback can significantly degrade the peformance of preference-based learning (Lee et al., 2021b). To filter out inaccurate pseudo-labels, we only use unlabeled samples for training when the confidence of the predictor is higher than a pre-defined threshold (Rosenberg et al., 2005). Then the reward model $r_\psi$ is optimized by minimizing the following objective:

$$\mathcal{L}^{\mathrm{SSL}} = \mathop{\mathbb{E}}_{\substack{(\sigma_l^0, \sigma_l^1, y) \sim \mathcal{D}_l, \\ (\sigma_u^0, \sigma_u^1) \sim \mathcal{D}_u}} \left[ \mathcal{L}^{\mathrm{Reward}}(\sigma_l^0, \sigma_l^1, y) + \lambda \cdot \mathcal{L}^{\mathrm{Reward}}(\sigma_u^0, \sigma_u^1, \widehat{y}) \cdot \mathbb{1}(P_\psi[\sigma_u^{k^*} \succ \sigma_u^{1-k^*}] > \tau) \right], \tag{3}$$

where $k^* = \arg\max_{j \in \{0,1\}} \widehat{y}(j)$ is an index of the preferred segment from the pseudo-label, $\lambda$ is a hyperparameter that balances the losses, and $\tau$ is a confidence threshold. Training with the pseudo-labels encourages the model to output more confident predictions on unlabeled samples. This can be seen as a form of entropy minimization (Grandvalet & Bengio, 2004), which is essential to the success of recent SSL methods (Berthelot et al., 2019; 2020). The entropy minimization can improve the reward learning by forcing the preference predictor to be low-entropy (i.e., high-confidence) on

---

[1]The unlabeled dataset $\mathcal{D}_u$ is not constrained to a fixed size since one can collect those unlabeled samples flexibly by sampling arbitrary pairs of experiences from the buffer.

---

**Algorithm 2** `TDA`: Temporal data augmentation for reward learning

---

**Require:** Minimum and maximum length $H_{\min}$ and $H_{\max}$, respectively, for cropping
**Require:** Pair of segments $(\sigma^0, \sigma^1)$ with length $H$
1: $\sigma^0 = \{(\mathbf{s}_0^0, \mathbf{a}_0^0), ..., (\mathbf{s}_{H-1}^0, \mathbf{a}_{H-1}^0)\}$
2: $\sigma^1 = \{(\mathbf{s}_0^1, \mathbf{a}_0^1), ..., (\mathbf{s}_{H-1}^1, \mathbf{a}_{H-1}^1)\}$
3: Sample $H'$ from a range of $[H_{\min}, H_{\max}]$
4: Sample $k_0, k_1$ from a range of $[0, H - H']$
5: // RANDOMLY CROP A SEQUENCE WITH LENGTH $H'$
6: $\widehat{\sigma}^0 \leftarrow \{(\mathbf{s}_{k_0}^0, \mathbf{a}_{k_0}^0), ..., (\mathbf{s}_{k_0+H'-1}^0, \mathbf{a}_{k_0+H'-1}^0)\}$
7: $\widehat{\sigma}^1 \leftarrow \{(\mathbf{s}_{k_1}^1, \mathbf{a}_{k_1}^1), ..., (\mathbf{s}_{k_1+H'-1}^1, \mathbf{a}_{k_1+H'-1}^1)\}$
8: Return $(\widehat{\sigma}^0, \widehat{\sigma}^1)$

---

unlabeled samples. During training, we sample a larger minibatch of unlabeled samples than labeled ones by a factor of $\mu$ following (Sohn et al., 2020), since unlabeled samples with low confidence are dropped within minibatch.

## 4.2 TEMPORAL DATA AUGMENTATION FOR REWARD LEARNING

To further improve the feedback-efficiency in preference-based RL, we propose a new data augmentation technique specially designed for reward learning. Specifically, for a given two segments and preference $(\sigma^0, \sigma^1, y)$, we generate augmented segments $(\widehat{\sigma}^0, \widehat{\sigma}^1, y)$ by cropping the subsequence from each segment (see Algorithm 2 for more details).[2] Then, we utilize augmented samples $(\widehat{\sigma}^0, \widehat{\sigma}^1)$ to optimize the cross-entropy loss in (3). The intuition behind the augmentation is that for a given pair of behavior clips, the human teacher may keep their relative preferences for slightly shifted or resized versions of them. In the context of SSL, data augmentation is also related to consistency regularization (Xie et al., 2019; Sohn et al., 2020) approaches that train the model to output similar predictions on augmented versions of the same sample. Namely, this *temporal cropping* method enables our framework can also enjoy the benefits of consistency regularization.

## 5 EXPERIMENTS

We design our experiments to investigate the following:

- ○ How does SURF improve the existing preference-based RL method in terms of feedback efficiency?
- ○ What is the contribution of each of the proposed components in SURF?
- ○ How does the number of queries affect the performance of SURF?
- ○ Is temporal cropping better than existing state-based data augmentation methods in terms of feedback efficiency?
- ○ Can SURF improve the performance of preference-based RL methods when we operate on high-dimensional and partially observable inputs?

## 5.1 SETUPS

We evaluate SURF on several complex robotic manipulation and locomotion tasks from Meta-world (Yu et al., 2020) and DeepMind Control Suite (DMControl; Tassa et al. 2018; 2020), respectively. Similar to prior works (Christiano et al., 2017; Lee et al., 2021a;b), in order to systemically evaluate the performance, we consider a scripted teacher that provides preferences between two trajectory segments to the agent according to the underlying reward function.[3] Since preferences of the scripted teacher exactly reflects ground truth reward of the environment, one can evaluate the algorithms quantitatively by measuring the true return.

---

[2]The length of the cropped segment is generated randomly across the batch but the same for segment pairs, because the preference predictor uses the accumulated sum of the reward over time.
[3]While utilizing preferences from the human teacher is ideal, this makes hard to evaluate algorithms quantitatively and quickly.

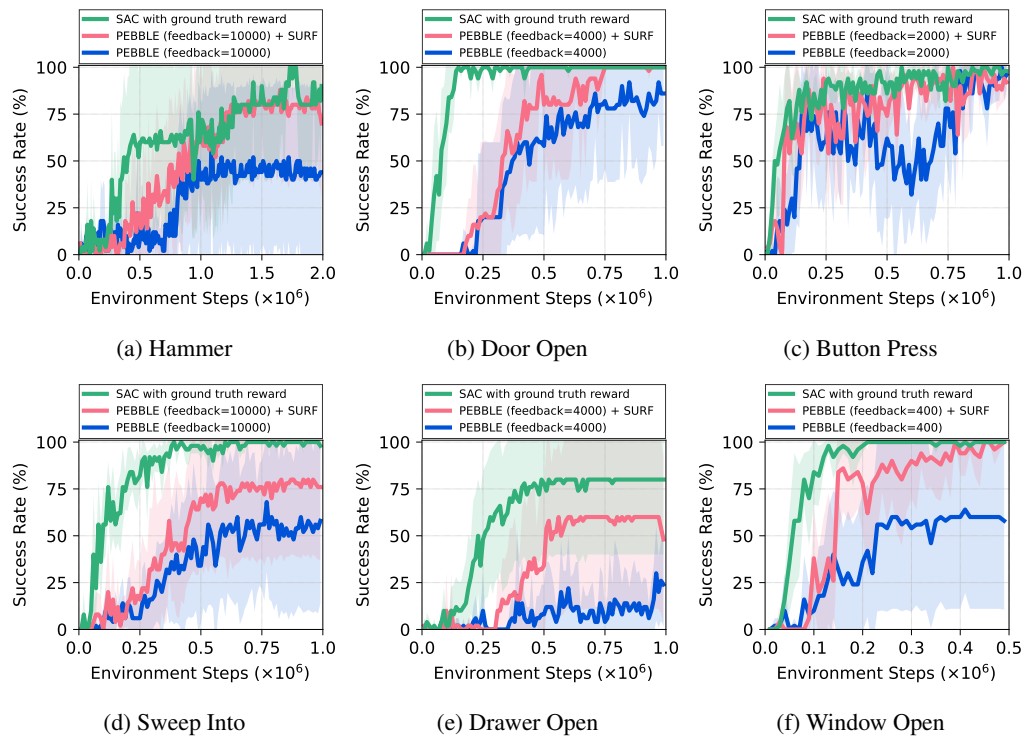

Figure 2: Learning curves on robotic manipulation tasks as measured on the success rate. The solid line and shaded regions represent the mean and standard deviation, respectively, across five runs.

We remark that SURF can be combined with any preference-based RL algorithms by replacing the reward learning procedure of its backbone method. In our experiments, we choose state-of-the-art approach, PEBBLE (Lee et al., 2021a), as our backbone algorithm. Since PEBBLE utilizes SAC (Haarnoja et al., 2018) algorithm to learn the policy, we also compare to SAC using the ground truth reward directly, as an upper bound of PEBBLE and our method. We note that our goal is not to outperform SAC, but rather to perform closely using as few preference queries as possible.

**Implementation details of SURF**. For all experiments, we use the same hyperparameters used by the original SAC and PEBBLE algorithms, such as learning rate of neural networks and frequency of the feedback session. For query selection strategy, we use the disagreement-based sampling scheme, which selects queries with high uncertainty, i.e., ensemble disagreement (see Appendix B for more details). At each feedback session, we sample unlabeled samples as 10 times of labeled ones by uniform sampling scheme, unless otherwise noted. Although we only use such amount of unlabeled samples for time-efficient training, we note that one can utilize much more unlabeled samples as needed. For the hyperparameters of SURF, we fix the loss weight $\lambda = 1$, and unlabeled batch ratio $\mu = 4$ for all experiments, and use threshold parameter $\tau = 0.999$ for Window Open, Sweep Into, Cheetah tasks, and $\tau = 0.99$ for the others. We provide more experimental details in Appendix B.

**Extension to visual control tasks**. To further demonstrate the effectiveness of our method, we also provide experimental results on visual control tasks, where each observation is an image of $84 \times 84 \times 3$. Specifically, we choose DrQ-v2 (Yarats et al., 2022), a state-of-the-art pixel-based RL approach on DMControl, as a backbone algorithm for PEBBLE and SURF. Similar to experiments with state-based inputs, we also compare to DrQ-v2 using ground truth reward as an upper bound.

## 5.2 BENCHMARK TASKS WITH SCRIPTED TEACHERS

**Meta-world experiments**. Meta-world consists of 50 robotic manipulation tasks, which are designed for learning diverse manipulation skills. We consider six tasks from Meta-world, to investigate how SURF improves a preference-based learning method on a range of complex robotic manipulation tasks (see Figure 7 in Appendix B). Figure 2 shows the learning curves of SAC, PEBBLE

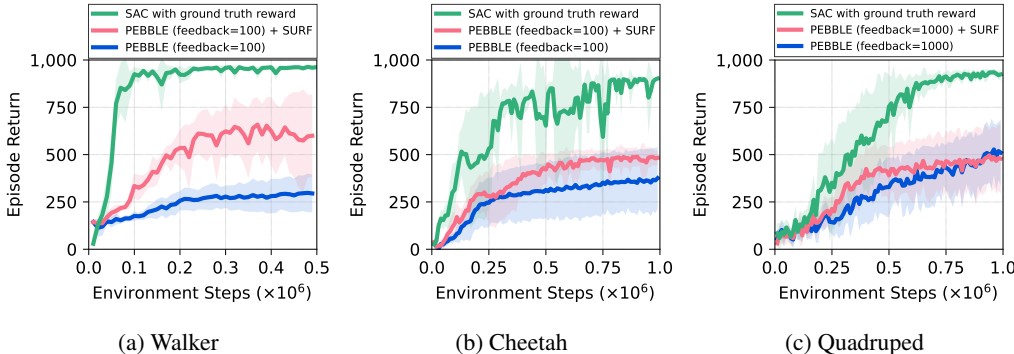

Figure 3: Learning curves on locomotion tasks as measured on the ground truth reward. The solid line and shaded regions represent the mean and standard deviation, respectively, across five runs.

and SURF (which combined with PEBBLE) on the manipulation tasks. In each task, PEBBLE and SURF utilize the same number of preference queries for fair comparison. As shown in the figure, SURF significantly improves the performance of PEBBLE given the same number of feedback on all tasks we considered, and matches the performance of SAC using the ground truth reward on four tasks. For example, we find that when using 400 preference queries, SURF (red) reaches the same performance as SAC (green) while PEBBLE (blue) is far behind to SAC on Window Open task. We also observe that SURF achieves similar performance to PEBBLE with much less labels. For example, to achieve comparable performance to SAC on Window Open task, PEBBLE needs 2,500 queries (reported in (Lee et al., 2021a)), requiring about 6 times more queries than SURF. These results demonstrate that SURF significantly reduces the feedback required to solve complex tasks.

**DMControl experiments**. For locomotion tasks, we choose three complex environments from DMControl: Walker-walk, Cheetah-run, and Quadruped-walk. Figure 3 shows the learning curves of the algorithms with same number of queries. We find that using a budget of 100 or 1,000 queries (which takes only few human minutes), SURF (red) could significantly improve the performance of PEBBLE (blue). These results again demonstrate that that SURF improves the feedback-efficiency of preference-based RL methods on a variety of complex tasks.

## 5.3 ABLATION STUDY

**Component analysis**. To evaluate the effect of each technique in SURF individually, we incrementally apply semi-supervised learning (SSL) and temporal cropping (TC) to our backbone algorithm, PEBBLE. Figure 4a shows the learning curves of SURF on Walker-walk task with 100 queries. We observe that leveraging unlabeled samples via pseudo-labeling (green) significantly improves PEBBLE, in terms of both sample-efficiency and asymptotic performance, while standard PEBBLE (blue) suffers from lack of supervision. In addition, both supervised (blue) and semi-supervised (green) reward learning are further improved by additionally utilizing temporal cropping (purple and red, respectively). This implies that our augmentation method improves label-efficiency by generating diverse behaviors share the same labels. Also, the results show that the key components of SURF are both effective, and their combination is essential to our method's success. We also provide extensive ablation studies on Meta-world, which show similar tendencies in Appendix C.

**Effects of query size**. To investigate how the number of queries affects the performance of SURF, we evaluate the performance of SURF with a varying number of queries $N \in \{50, 100, 200, 400\}$. As shown in Figure 4b, SURF (solid lines) consistently improves the performance of PEBBLE (dotted lines) across a wide range of query sizes. The gain from SURF becomes even more significant in the extreme label-scarce scenarios, i.e., $N \in \{50, 100\}$.

**Comparison to other augmentation for state-based inputs**. To demonstrate that temporal cropping can enduce significant improvements for reward learning, we compare our method to other augmentation methods for state-based inputs. We consider random amplitude scaling (RAS) and adding Gaussian noise (GN) proposed in Laskin et al. (2020) as our baselines. RAS multiplies an uniform random variable $z$ to the state, i.e., $\widehat{\mathbf{s}} = \mathbf{s} \cdot z$, where $z \sim \text{Unif}[\alpha, \beta]$, and GN adds

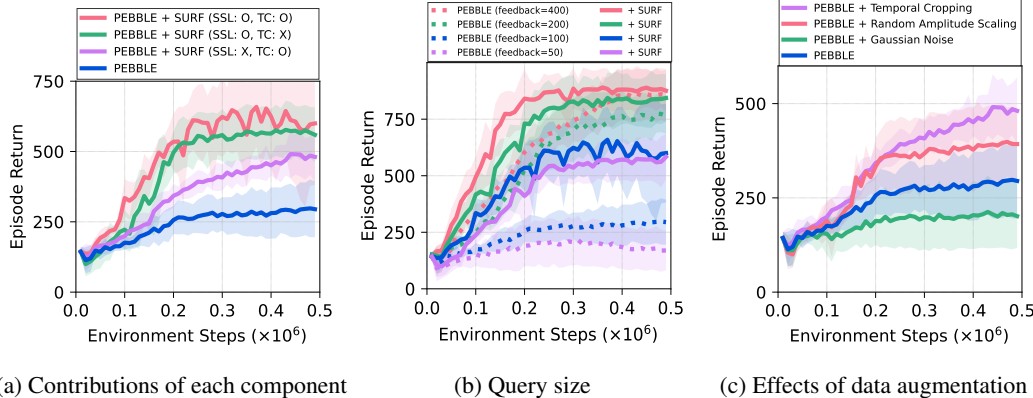

(a) Contributions of each component    (b) Query size    (c) Effects of data augmentation

Figure 4: Ablation study on Walker-walk. (a) Contribution of each technique in SURF, i.e., semi-supervised learning (SSL) and temporal cropping (TC). (b) Effects of query size. (c) Comparison of augmentation methods. The results show the mean and standard deviation averaged over five runs.

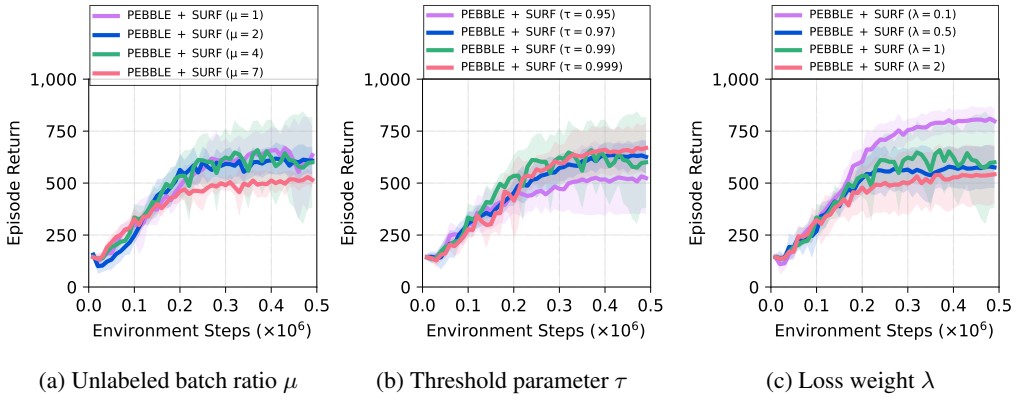

(a) Unlabeled batch ratio $\mu$    (b) Threshold parameter $\tau$    (c) Loss weight $\lambda$

Figure 5: Hyperparameter analysis on Walker-walk using 100 preference queries. The results show the mean and standard deviation averaged over five runs.

a multivariate Gaussian random variable $\mathbf{z}$ to the state, i.e., $\widehat{\mathbf{s}} = \mathbf{s} + \mathbf{z}$, where $\mathbf{z} \sim \mathcal{N}(0, I)$. As proposed in Laskin et al. (2020), we apply these methods consistently along the time dimension, and choose the parameters for RAS as $\alpha = 0.8$, $\beta = 1.2$. Specifically, for a given segment $\sigma = (\{(\mathbf{s}_k, \mathbf{a}_k), ..., (\mathbf{s}_{k+H-1}, \mathbf{a}_{k+H-1})\})$, we obtain the augmented sample $\widehat{\sigma}$ by perturbing each state along the segment, i.e., $\widehat{\sigma} = (\{(\widehat{\mathbf{s}}_k, \mathbf{a}_k), ..., (\widehat{\mathbf{s}}_{k+H-1}, \mathbf{a}_{k+H-1})\})$. In Figure 4c, we plot the learning curves of PEBBLE with various data augmentations on Walker-walk task with 100 queries. We observe that RAS improves the performance of PEBBLE, but temporal cropping still outperforms these two methods. GN degrades the performance, possibly due to the noisy inputs. Since RAS is an orthogonal approach to augment state-based inputs, one can integrate them with our method to further improve the performance. This may be an interesting future direction for addressing feedback-efficiency in preference-based RL (see Appendix C).

**Effects of hyperparameters of SURF**. We investigate how the hyperparameters of SURF affect the performance of preference-based RL. In Figure 5, we plot the learning curve of SURF with different set of hyperparameters: (a) unlabeled batch ratio $\mu \in \{1, 2, 4, 7\}$, (b) threshold parameter $\tau \in \{0.95, 0.97, 0.99, 0.999\}$, and (c) loss weight $\lambda \in \{0.1, 0.5, 1, 2\}$, respectively. First, we observe that SURF is quite robust on $\mu$, but the performance slightly drops with a large batch size $\mu = 7$. We expect that this is because a large batch size makes the reward model overfit to unlabeled data. We also observe that SURF is also robust on the threshold $\tau$, except for the smallest value of 0.95. Because there are only two classes in tasks, the optimal threshold could larger than the value typically used in previous SSL methods (Sohn et al., 2020), i.e., 0.95. In the case of the loss weight $\lambda$, tuning this parameter brings more improvements than other hyperparameters. Although

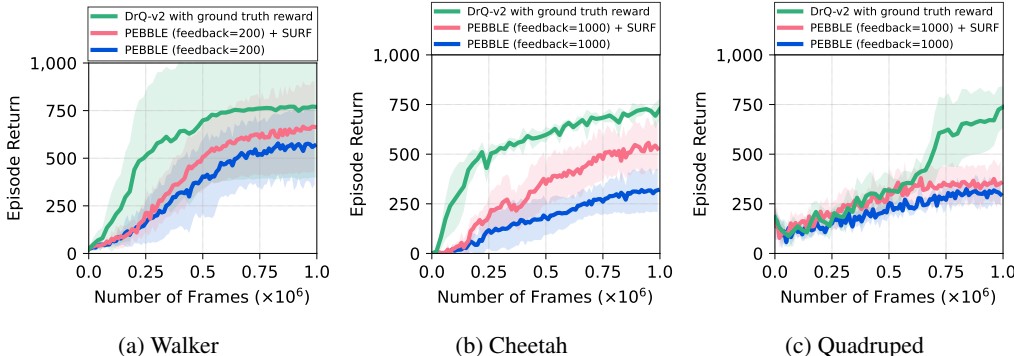

Figure 6: Learning curves on locomotion tasks with pixel-based inputs as measured on the ground truth reward. The solid line and shaded regions represent the mean and standard deviation, respectively, across five runs.

we use a simple choice, i.e., $\lambda = 1$, in our experiments, more tuning $\lambda$ would further improve the performance of our method.

## 5.4 Experiments on visual control tasks

Figure 6 shows the learning curve of DrQ-v2, PEBBLE, and SURF with the same number of queries. We observe that SURF (red) significantly improves the performance of PEBBLE (blue). In particular, SURF achieves comparable performance to DrQ-v2 (green) with ground truth reward in Walker-walk, only using a budget of 200 queries. These results demonstrate that SURF could also improve the performance with image observations. We remark that our temporal cropping augmentation can be combined with any existing image augmentation methods, which would be an interesting future direction to explore.

## 6 Discussion

In this work, we present SURF, a semi-supervised reward learning algorithm with data augmentation for preference-based RL. First, in order to utilize an unlimited number of unlabeled data, we utilize pseudo-labeling on confident samples. Also, to enforce consistencies to the reward function, we propose a new data augmentation method called temporal cropping. Our experiments demonstrate that SURF significantly improves feedback-efficiency of current state-of-the-art method on a variety of complex robotic manipulation and locomotion tasks. We believe that SURF can scale up deep RL to more diverse and challenging domains by making preference-based learning more tractable.

An interesting future direction is to extend state-based inputs to partially-observable or high-dimensional inputs, e.g., pixels. One can expect that representation learning based on unlabeled samples and data augmentation (Chen et al., 2020a; Grill et al., 2020) is crucial to handle such inputs. We think that our investigations on leveraging unlabeled samples and data augmentation would be useful in representation learning for preference-based RL.

**Ethics statement**. Preference-based RL can align RL agents with the teacher's preferences, which enables us to apply RL to diverse problems and obtain strong AI. However, there could be possible negative impacts if a malicious user corrupts the preferences to teach the agent harmful behaviors. Since we have proposed a method that makes preference-based RL algorithms more feedback-efficiently, our method may reduce the efforts for teaching not only the desirable behaviors, but also such bad behaviors. For this reason, in addition to developing algorithms for better performance and efficiency, it is also important to consider safe adaptation in the real world.

**Reproducibility statement**. We describe the implementation details of SURF in Appendix B, and also provide our source code in the supplementary material.

ACKNOWLEDGEMENTS AND DISCLOSURE OF FUNDING

This work was supported by Samsung Electronics Co., Ltd (IO201211-08107-01), OpenPhilosophy, and Institute of Information & Communications Technology Planning & Evaluation (IITP) grant funded by the Korea government (MSIT) (No.2019-0-00075, Artificial Intelligence Graduate School Program (KAIST)). We would like to thank Junsu Kim and anonymous reviewers for providing helpful feedbacks and suggestions in improving our paper.

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

## A PEBBLE ALGORITHM

A state-of-the-art preference-based RL algorithm, PEBBLE (Lee et al., 2021a), consists of two main components: unsupervised pre-training and relabeling experiences. To collect diverse experience, PEBBLE pre-trains the policy by using intrinsic motivation (Oudeyer et al., 2007; Schmidhuber, 2010) in the beginning of training. Specifically, PEBBLE optimizes the policy to maximize the state entropy $\mathcal{H}(\mathbf{s}) = -\mathbb{E}_{\mathbf{s} \sim p(\mathbf{s})} [\log p(\mathbf{s})]$ to efficiently explore the environment. Then PEBBLE learns the policy by using the state-of-the-art off-policy RL algorithm, SAC (Haarnoja et al., 2018). Since the learning process of off-policy algorithms with a non-stationary reward function can be unstable, PEBBLE stabilizes the learning process by relabeling all experiences in the buffer when the reward model is updated.

## B EXPERIMENTAL DETAILS

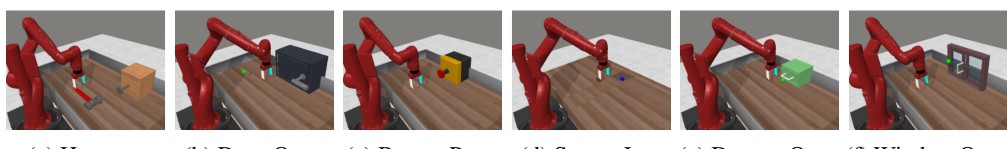

(a) Hammer    (b) Door Open    (c) Button Press    (d) Sweep Into    (e) Drawer Open (f) Window Open

Figure 7: Rendered images of robotic manipulation tasks from Meta-world. Our goal is learning various locomotion and manipulation skills using preferences from a teacher.

**Training details**. We choose PEBBLE (Lee et al., 2021a) as a backbone algorithm of SURF, and use the hyperparameters in Table 1 for both PEBBLE and our method. For the reward model, we use a three-layer MLP with 256 hidden units and leaky ReLU activation. Following the implementation of PEBBLE (Lee et al., 2021a), we use an ensemble of three reward models and bound the output to $[-1, 1]$ using tanh function. Each model is trained by minimizing the cross-entropy loss using ADAM optimizer (Kingma & Ba, 2015) with the learning rate of 0.0003. For semi-supervised learning and data augmentation of SURF, we use hyperparameters in Table 2.

**Sampling schemes**. In preference-based RL methods, informative query sampling (Biyik & Sadigh, 2018; Biyik et al., 2020; Sadigh et al., 2017) has been adopted for improving the feedback-efficiency. For all experiments of PEBBLE and SURF, we use the disagreement-based sampling (Christiano et al., 2017) to choose queries for labeling: we first uniformly sample the initial batch of segments, and select $N_{\texttt{query}}$ pairs of segments[4] with high uncertainty based on the variance across ensemble of preference predictors $\{P_{\psi_i}[\sigma^1 \succ \sigma^0]\}_{i=1}^{N_{\texttt{en}}}$. Note that we use uniform sampling scheme for unlabeled samples, because the number of unlabeled samples are not limited. At each feedback session, we sample unlabeled samples as 10 times of labeled ones if the maximum budget of feedback is equal or larger than 1,000, and otherwise we sample unlabeled samples as 100 times of labeled ones.

Table 1: Hyperparameters of PEBBLE.

| Hyperparameter | Value | Hyperparameter | Value |
|---|---|---|---|
| Initial temperature | 0.1 | Hidden units per each layer | 1024 (DMControl), 256 (Meta-world) |
| Length of segment | 50 | # of layers | 2 (DMControl), 3 (Meta-world) |
| Learning rate | 0.0003 (Meta-world) | Batch Size | 1024 (DMControl), 512 (Meta-world) |
| | 0.0005 (Walker, Cheetah) | Optimizer | Adam (Kingma & Ba, 2015) |
| | 0.0001 (Quadruped) | | |
| Critic target update freq | 2 | Critic EMA $\tau$ | 0.005 |
| $(\beta_1, \beta_2)$ | (0.9, 0.999) | Discount $\bar{\gamma}$ | 0.99 |
| Frequency of feedback | 5000 (Meta-world) | Maximum budget / | 1000/100, 100/10 (DMControl) |
| | 20000 (Walker, Cheetah) | # of queries per session | 10000/50, 4000/20 (Meta-world) |
| | 30000 (Quadruped) | | 2000/25, 400/10 (Meta-world) |
| # of ensemble models $N_{\texttt{en}}$ | 3 | # of pre-training steps | 10000 |

---

[4]We select 10% of the initial batch.

Table 2: Hyperparameters of SURF with state-based inputs.

| Hyperparameter | Value |
|---|---|
| Unlabeled batch ratio $\mu$ | 4 |
| Threshold $\tau$ | 0.999 (Window Open, Sweep Into, Cheetah) |
| | 0.99 (others) |
| Loss weight $\lambda$ | 1 |
| Min/Max length of cropped segment $[H_{\min}, H_{\max}]$ | $[45, 55]$ |
| Segment length before cropping | 60 |

**Hyperparameters of DrQ-v2**. We use the same encoder architecture and hyperparameters as in DrQ-v2 (Yarats et al., 2022). The full list of hyperparameters of DrQ-v2 is presented in Table 3. For semi-supervised learning and data augmentation of SURF with pixel-based inputs, we use hyperparameters in Table 4.

Table 3: A set of hyperparameters used in our experiments.

| Parameter | Setting |
|---|---|
| Replay buffer capacity | $10^6$ |
| Action repeat | 2 |
| Seed frames | 4000 |
| Exploration steps | 2000 |
| $n$-step returns | 1 (Walker), 3 (Cheetah / Quadruped) |
| Mini-batch size | 512 (Walker), 256 (Cheetah / Quadruped) |
| Discount $\gamma$ | 0.99 |
| Optimizer | Adam |
| Learning rate | $10^{-4}$ |
| Agent update frequency | 2 |
| Critic Q-function soft-update rate $\tau$ | 0.01 |
| Features dim. | 50 |
| Hidden dim. | 1024 |
| Exploration stddev. clip | 0.3 |
| Exploration stddev. schedule | Walker: $\text{linear}(1.0, 0.1, 100000)$ |
| | Cheetah / Quadruped: $\text{linear}(1.0, 0.1, 500000)$ |

Table 4: Hyperparameters of SURF with pixel-based inputs.

| Hyperparameter | Value |
|---|---|
| Unlabeled batch ratio $\mu$ | 1 |
| Threshold $\tau$ | 0.99 |
| Loss weight $\lambda$ | 1 (Cheetah), 0.1 (others) |
| Min/Max length of cropped segment $[H_{\min}, H_{\max}]$ | $[45, 55]$ (Cheetah), $[48, 52]$ (others) |
| Segment length before cropping | 60 (Cheetah), 54 (others) |

**Implementation**. We implement SURF using the publicly released implementation repository of the PEBBLE algorithm (https://github.com/pokaxpoka/B_Pref) with a full list of hyperparameters in Table 1. For visual control tasks, we implement PEBBLE and SURF using the official code of DrQ-v2 (https://github.com/facebookresearch/drqv2) with hyperparameters in Table 3. Note that DMControl environment depends on the MuJoCo simulator (Todorov et al., 2012), which is a commercial software. We follow the standard evaluation protocol for the locomotion tasks from DMControl. For robotic manipulation tasks from Meta-world, we measure the task success rate as defined by the authors. For each run of experiments, we utilize one Nvidia RTX 2080 Ti GPU and 4 CPU cores for training.

## C  ADDITIONAL EXPERIMENTAL RESULTS

**Ablation study on Meta-world**. We provide additional experimental results for component analysis on Meta-world (Yu et al., 2020). To evaluate the effect of each technique in SURF individually, we incrementally apply semi-supervised learning (SSL) and temporal cropping (TC) to our backbone

algorithm, PEBBLE. Figure 8a and 8b show the learning curves of SURF on Window Open with 400 queries and Hammer with 10,000 queries, respectively. We observe that both semi-supervised learning (green) and data augmentation (purple) improve the baseline of PEBBLE (blue). Also, applying both of them further improves the performance (red). This shows that the key components of SURF are both effective.

**Applying temporal cropping with other augmentations**. In Section 5.3, we compared our method to random amplitude scaling (RAS) and adding Gaussian noise (GN) proposed in Laskin et al. (2020). To investigate if applying RAS or GN with the temporal cropping (TC) further improve the performance, we provide experimental results for the joint usage of the augmentations. In Figure 9a, we plot the learning curves of PEBBLE with various data augmentations on Walker-walk with 100 queries. We observe that the naive combination of two augmentations, i.e., RAS + TC and GN + TC, do not further improve the performance. Investigating how to combine several data augmentation methods for reward learning would be an interesting future direction.

**Effects of augmentation intensity**. To investigate how does augmentation intensity affects the RL performance, we provide additional experimental results. In Walker-walk with 100 queries, we apply the temporal cropping to PEBBLE, with a varying cropping range, i.e., $(H_{\mathtt{max}} - H_{\mathtt{min}})/2$, from an array of $[2, 5, 10, 20]$. Figure 9b shows that temporal cropping consistently improves the performance of the PEBBLE. Although the performance with a large cropping range of 20 slightly underperforms the performance with 10, these results show that our augmentation method is quite robust to the choice of hyperparameters.

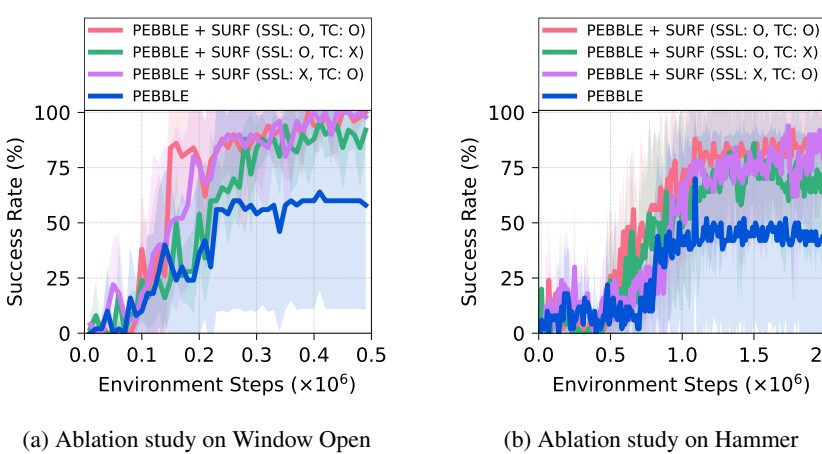

(a) Ablation study on Window Open      (b) Ablation study on Hammer

Figure 8: Contribution of each technique in SURF, i.e., semi-supervised learning (SSL) and temporal cropping (TC), in (a) Window Open, and (b) Hammer. The results show the mean and standard deviation averaged over five runs.

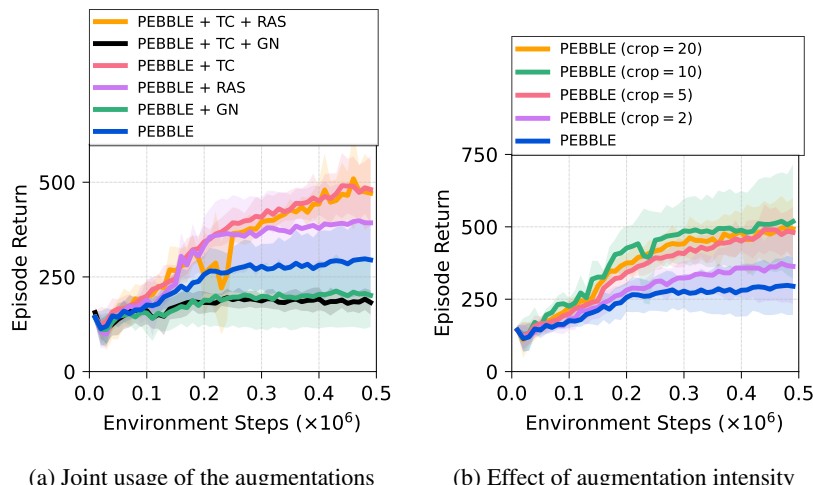

(a) Joint usage of the augmentations      (b) Effect of augmentation intensity

Figure 9: Analysis of the augmentation methods on Walker-walk task with 100 queries. (a) Comparison of the augmentation methods, i.e., applying random amplitude scaling (RAS) or Gaussian noise (GN) with temporal cropping (TC). (b) Comparison of the performance of PEBBLE with varying augmentation intensity, i.e., cropping range. The results show the mean and standard deviation averaged over five runs.

