# OpenReview forum: "SURF: Semi-supervised Reward Learning with Data Augmentation for Feedback-efficient Preference-based Reinforcement Learning"
_ICLR.cc/2022/Conference — ICLR 2022 Poster_

### Official Review · Reviewer_JjVp · 2021-10-29

**Correctness:** 3
**Technical Novelty And Significance:** 3
**Empirical Novelty And Significance:** 3
**Recommendation:** 6
**Confidence:** 4

**Main Review:**

The authors use a preference-based reinforcement learning method, which is based on a surrogate reward model. The surrogate reward model is learned by phrasing the preference learning problem as classification over pairwise comparisons. The semi-supervised learning approach uses the preference classifier for deriving high-confidence pseudo labels, that can be used as additional, supervised input data.
Both are well known and established principles, making the proposed method a sound contribution. However, as the bridge between classification and PBRL, as well as classification based SSL, are already well known, this is a straight forward application of known methods to a new problem domain, limiting the novelty. The explanation and formalization of this section is good and clear to understand, with one small exception: Two different preference notations are used: (0,1) and >. The first notation is also used for introducing an equivalence relation, which is not used otherwise. This should be unified to the more common > notation (for further preference relation symbols, see https://www.jmlr.org/papers/volume18/16-634/16-634.pdf). An additional, potential improvement is possible by using the tau threshold for defining a confidence cutoff between preference and equivalence relation. However, both are minor issues and do not impact clarity.

Second, the authors introduce a novel data augmentation scheme, by selecting random subsequences out of the observed trajectories, which is also clearly understandable. This method has the implicit assumption, that the reason (states) for the observed preference are still part of the shorter sequence. Especially, in sparse reward domains, this is not a given.

The main evaluation is good, and sufficiently diverse. Advantages of the proposed method are clearly visible. Adding some non-robotics (potentially high dimensional) domains could improve the paper further, but this was also clearly stated by the authors. The given details (including appendix), should be sufficient for recreating the experiment setup.

The ablation study is a welcome addition and the authors covered all relevant aspects, meaning not just the two parts of the introduced method, but also several other factors (and hyperparameters) that can potentially have a substantial impact on the results. However, the ablation study is limited to a single domain, limiting the generality of the derived statements. Especially, i consider the advantage of TDA not sufficiently supported, considering that walker-walk is not a sparse reward domain.

**Summary Of The Paper:**

The authors introduce a method for combining semi-supervised learning with preference learning for reinforcement learning. The method uses pseudo labels, derived from a surrogate preference model. Furthermore, a data augmentation scheme for this setting is introduced, which is based on cropped sequences. Evaluation is performed on a range of robotics tasks, including ablation studies for several hyperparameters and the two, distinct parts of the introduced method.

**Summary Of The Review:**

The proposed method is sound and interesting, but of limited novelty. The support for the full method is good, but without an extended ablation study, the support for TDA is not sufficient. The method is likely significant within the area of PBRL, but but not for a broad audience. The paper is clearly written and can be easily understood.

---

> ### Author Response · Authors · 2021-11-19
> **Response to Reviewer JjVp**
>
> Dear Reviewer JjVp,
>
> We sincerely appreciate your valuable and insightful comments. We found them extremely helpful for improving our manuscript. We have updated our revision based on your comments and colored by red. We address each comment in detail, one by one below.
>
> ---
>
> **Q1. Limited novelty.**
>
> **A1.**  We would like to remark that our two contributions, i.e., (i) an introduction of new data augmentation for preference-based RL and (ii) a novel combination of semi-supervised learning and the proposed data augmentation, are not straightforward applications; they were usually not utilized in preference-based RL literature, and we first show that data augmentations and semi-supervised learning are useful in the preference-based RL setting. In particular, we think our effective, yet simple, semi-supervised learning technique can be an important milestone in the preference-based RL literature, as an unlimited number of unlabeled data can be easily obtained with no additional cost, i.e., from past experiences stored in the buffer. We emphasized our contributions in the introduction of the revised draft to make these points clearer.
>
> ---
>
> **Q2. Does the assumption of the augmentation hold in the sparse reward setup?**
>
> **A2.**  We expect our assumption that the preference would be consistent on the temporally cropped segments still holds in many sparse-reward setups. In a sparse-reward environment, even for trajectory segments with zero rewards, humans can give preferences over them by comparing which segment is better for solving the task, e.g., approaching the goal. Then one can expect that the teacher maintains the preference over their consecutive subsequences, so our assumption may hold in such cases.
>
> For example, Meta-world tasks in our experiments (Section 5.2) become sparse reward setups if we use the indicator for task completion as a true reward. Instead of learning this underlying sparse reward, we train a (dense) "pseudo-reward" using the preferences of the scripted teacher, which approximates the human. In additional experimental results of ablation study for Meta-world (see our response A3 below), we observed that our augmentation method successfully improves the performance, which implies that our assumption holds in such a sparse reward setup.
>
> ---
>
> **Q3. Extended ablation study.**
>
> **A3.**  Thanks for the valuable suggestion. We conducted additional experimental results for component analysis on Window Open and Hammer of Meta-world, in Appendix C of the revised draft. We observe that both semi-supervised learning and data augmentation improve the baseline of supervised learning, and applying both of them further improves the performance. This again shows that the key components of SURF are both effective. We will include relevant discussion and more experimental results in the final draft.
>
> ---
>
> **Q4. Description for preference notations.**
>
> **A4.**  Thank you for pointing this out. We followed the notation of our previous related work [1, 2, 3], however, we agree that unifying the preference notations would be more clear to understand. We have unified these notations in the revised draft.
>
> ---
>
> **References**
>
> [1] Paul F Christiano, Jan Leike, Tom Brown, Miljan Martic, Shane Legg, and Dario Amodei. Deep reinforcement learning from human preferences. In Advances in Neural Information Processing Systems, 2017.
>
> [2] Kimin Lee, Laura Smith, and Pieter Abbeel. Pebble: Feedback-efficient interactive reinforcement learning via relabeling experience and unsupervised pre-training. In International Conference on Machine Learning, 2021.
>
> [3] Kimin Lee, Laura Smith, Anca Dragan, and Pieter Abbeel. B-pref: Benchmarking preference-based reinforcement learning. In Thirty-fifth Conference on Neural Information Processing Systems Datasets and Benchmarks Track (Round 1), 2021.

---

> > ### Comment · Reviewer_JjVp · 2021-11-22
> > **Response to Authors**
> >
> > Thanks for the extensive rebuttal and the introduced changes. Understandability and correctness has been improved, however, i am still doubtful concerning TC. I think that it will be easy to construct counterexample domains, where this will fail. However, i acknowledge the additional results, showing that TC seems to at least work reasonable in several domains. Although, the new results make me wonder, why the single modifications are nearly as good as the joint approach, but this is only something to consider for further work. Resultingly, i raised my evaluation slightly, but higher scores are still hampered by the novelty and TC issues.

---

> > > ### Author Response · Authors · 2021-11-23
> > > **Response about the TC issue**
> > >
> > > We are happy to hear that our response and additional results improve our paper, and thank you again for the valuable suggestions and comments.
> > >
> > > Regarding the temporal cropping augmentation, we agree that there could be some counterexample domains, but we believe that our augmentation is still effective in a wide range of domains, as we discussed in A2 of the response above (and A3 for Reviewer qrsj).
> > >
> > > To convince you further why this is not a weakness of our work, consider the random cropping augmentation which is widely-used in the computer vision domain. One can find or construct easily some counterexample images such that it discards semantically important information, i.e., hurt the model performance. Another example is color jittering, which is risky to use for color-sensitive domains. In fact, we think that most augmentation techniques have such counterexamples, despite their popularity and importance.
> > >
> > > We will include relevant discussions and results in the final draft.
> > >
> > > If you have any remaining suggestions or concerns, please let us know!
> > >
> > > Thank you,
> > >
> > > Authors.

---

> ### Author Response · Authors · 2021-11-22
> **A gentle reminder**
>
> Dear Reviewer JjVp,
>
> Thank you for your time and efforts in reviewing our paper.
>
> We kindly remind that the discussion period will end in 30 hours or so.
>
> We sincerely hope that our response and results of the supporting experiments successfully clarify your concerns.
>
> We just wonder whether we could have the last chance to address your further concerns or questions (if you have any).
>
> Thank you very much!
>
> Authors

---

### Official Review · Reviewer_cS2w · 2021-11-01

**Correctness:** 4
**Technical Novelty And Significance:** 3
**Empirical Novelty And Significance:** 3
**Recommendation:** 6
**Confidence:** 3

**Main Review:**

This paper aims to tackle an important problem in reinforcement learning, i.e., how to learn a policy without accessing a fine-grained reward function. More specifically, this paper focuses on preference-based RL that uses trajectory-wise comparison queries to guide the learning procedure. The paper is well organized and easy to follow. The proposed method is motivated by the success of data augmentation in the supervised learning area. It is interesting to see a connection between reinforcement learning and supervised learning.

Some questions:

- As shown in figure 6(b), the pseudo-labeling technique requires the hyper-parameter \tau to be a very large value. Why do we need to train our reward model on these sample pairs with extremely high confidence? Please correct me if I misunderstand anything. The loss computed by these high-confidence samples should be very small. Why it can provide a significant improvement on the final performance?

- RAS and GN presented in Figure 6(c) should be orthogonal to Temporal Cropping. I am curious whether we can further improve the performance by applying RAS and/or GN upon SURF.

- The ablation study is only conducted in one specific environment. It would be helpful to repeat these experiments in other tasks.

Minor comments:

- In section 3, regarding the notations of $\mathcal{L}^{CE}$, the terms $y(0)$ and $y(1)$ have not been defined.

- Algorithm 1 (TDA) is presented in page 4, which is far from section 4.2. It might be better to reverse the order of Algorithm 1 and 2 for presentation.


**Summary Of The Paper:**

This paper proposes two data-augmentation techniques to improve the query efficiency of preference-based RL. (1) Pseudo-labeling leverages unlabeled data by using high-confidence predictions as labels. (2) Temporal cropping augmentation generates imaginary comparisons by cropping trajectories.

The proposed augmentation can significantly improve the query efficiency of preference-based RL in a variety of benchmark tasks, including DMControl and Meta-World. The performance of SURF is comparable to dense-reward SAC while only accessing a few expert queries.

**Summary Of The Review:**

My recommendation score is marginally above the borderline. I like this paper overall but the discussions of ablation study need to be enriched.

The methodology introduced by this paper is inspired by related studies in supervised learning. I think it is quite important to understand why and how these techniques benefit RL algorithms.

---

> ### Author Response · Authors · 2021-11-19
> **Response to Reviewer cS2w**
>
> Dear Reviewer cS2w,
>
> We sincerely appreciate your valuable and insightful comments. We found them extremely helpful for improving our manuscript. We have updated our revision based on your comments and colored by red. We address each comment in detail, one by one below.
>
> ---
>
> **Q1. More discussion on pseudo-labeling with a high threshold.**
>
> **A1.**  The loss computed from high-confident samples could be small, but still gives a meaningful (and reliable) signal to the model because it encourages the model to produce even higher-confidence predictions on unlabeled samples, e.g., 0.99$\rightarrow$0.9999 (on the other hand, low-confident samples give high, yet wrong, signals to the model). During the training, this increases the number of unlabeled samples which have higher confidences than the threshold. Consequently, the model can utilize more unlabeled samples as training progresses, significantly outperforming the supervised learning that only uses labeled samples. Such pseudo-labeling with a high threshold has been well justified and popularly used in the semi-supervised learning literature [1, 2].
>
> ---
>
> **Q2. Applying RAS or GN upon temporal cropping.**
>
> **A2.**  Thanks for the valuable suggestion. We first evaluated the joint usage of RAS or GN with our temporal cropping in Appendix C of the revised draft, but the naive combination of two augmentations does not further improve the performance. Investigating how to combine several data augmentation methods for reward learning would be interesting to explore for future work.
>
> ---
>
> **Q3. Ablation study on other tasks.**
>
> **A3.**  Thanks for the valuable suggestion. We conducted additional experimental results for component analysis on Window Open and Hammer of Meta-world, in Appendix C of the revised draft. We observe that both semi-supervised learning and data augmentation improve the baseline of supervised learning, and applying both of them further improves the performance. This again shows that the key components of SURF are both effective. We will include relevant discussion and more experimental results in the final draft.
>
> ---
>
> **Q4. Detailed descriptions of the $\mathcal{L}^{\tt{CE}}$.**
>
> **A4.**  Thank you for pointing this out. First, $y$ is the teacher’s preference for given pair of segments $(\sigma^0, \sigma^1)$, i.e., $y=(1,0)$ means that $\sigma^0$ is preferable than $\sigma_1$ and vice versa. Then, $y(0)$ and $y(1)$ denote the value of each component of the preference $y$, i.e., $y = (y(0), y(1))$.
> For better clarity, we have updated the notations in the revised draft, as Reviewer JjVp suggested.
>
> ---
>
> **Q5. Order of Algorithms 1 and 2.**
>
> **A5.** Thanks for the suggestion. We have reordered Algorithms 1 and 2 in the revised draft for better presentation.
>
> ---
>
> **References**
>
> [1] D.-H. Lee. Pseudo-label: The simple and efficient semi-supervised learning method for deep neural networks. In ICML Workshop on Challenges in Representation Learning, 2013.
>
> [2] Kihyuk Sohn, David Berthelot, Chun-Liang Li, Zizhao Zhang, Nicholas Carlini, Ekin D Cubuk, Alex Kurakin, Han Zhang, and Colin Raffel. Fixmatch: Simplifying semi-supervised learning with consistency and confidence. In Advances in Neural Information Processing Systems, 2020.

---

> > ### Comment · Reviewer_cS2w · 2021-11-24
> > **Thanks for the thorough response**
> >
> > Thanks for the thorough response and additional experiments. My questions are well addressed.
> > I keep my vote for acceptance.

---

> ### Author Response · Authors · 2021-11-22
> **A gentle reminder**
>
> Dear Reviewer cS2w,
>
> Thank you for your time and efforts in reviewing our paper.
>
> We kindly remind that the discussion period will end in 30 hours or so.
>
> We sincerely hope that our response and results of the supporting experiments successfully clarify your concerns.
>
> We just wonder whether we could have the last chance to address your further concerns or questions (if you have any).
>
> Thank you very much!
>
> Authors

---

### Official Review · Reviewer_qrsj · 2021-11-02

**Correctness:** 3
**Technical Novelty And Significance:** 2
**Empirical Novelty And Significance:** 2
**Recommendation:** 6
**Confidence:** 4

**Main Review:**

Strengths:

s1) The paper proposes a semi-supervised reward learning pipeline aiming to reduce reward engineering efforts, which is an important topic.

s2) Extensive evaluation experiments are conducted to demonstrate the proposed approach.

Weaknesses:

w1) In Fig.1(b), the figure of temporal cropping seems incorrect. According to the description of algorithm 1, the red box should be within the blue box. In the second line, the first image in the red box is outside the blue box.

w2) In Sec 3. “Preference-based reinforcement learning,” “Then, we model a preference predictor using the reward function $\hat{r}_{\psi}$ following the Bradley-Terry model.” It’s not clear to the reviewer how the reward function is learned. “Specifically, given a dataset of preferences D, the reward function is updated by minimizing the binary cross-entropy loss.” Does the reward function have extra constraints? Assume $\psi$ is the reward function that satisfies the Eqn (1). It seems that there are infinite equivalent reward functions ($\psi$ + arbitrary constant number) if the length of segments (H) is the same.

w3) In Sec 4.1, the same question as the above question 2) w.r.t. the learning process of the reward function through the loss of Eqn(3). If multiple seeds are used, do these reward functions generate similar results on the samples?

w4) In Sec 4.2, “​​The intuition behind the augmentation is that for a given pair of behavior clips, the human teacher may keep their relative preferences for slightly shifted or resized versions of them.”  The assumption is too strong and does not generally hold. It will actually strengthen the paper if more analyses are provided. What would happen in situations when the assumption does not hold?



**Summary Of The Paper:**

This work introduces a semi-supervised reward learning approach to reduce the efforts of reward engineering, which contains two key components. The first is to produce artificial labels for these unlabeled samples leveraging the pseudo-labeling and the learned reference predictor. The second component is to crop consecutive subsequences for data augmentation. The proposed approach is tested on Meta-world and DMControl suites, and the results show that it significantly improves the performances.


**Summary Of The Review:**

This work introduces a semi-supervised reward learning approach to reduce the efforts of reward engineering. However, the technical novelty is limited. Some assumptions made by this work are too strong and may not hold. Some important descriptions (e.g., the reward function w2 and w3 in Main Review) are missing.

---

> ### Author Response · Authors · 2021-11-19
> **Response to Reviewer qrsj**
>
> Dear Reviewer qrsj,
>
> We sincerely appreciate your valuable and insightful comments. We found them extremely helpful for improving our manuscript. We have updated our revision based on your comments and colored by red. We address each comment in detail, one by one below.
>
> ---
>
> **Q1. Clarification of Figure 1(b).**
>
> **A1.**  Thank you for pointing this out. We have corrected Figure 1(b) in the revised draft.
> In addition, we clarify how our temporal cropping method works. In our experiments, we sampled a pair of segments with length 50, and query them to the teacher for labeling. After that, we stored those segments with extra margins of 5 on both left and right sides, i.e., we stored segments of length $H=60$. In the reward learning, we randomly choose the cropping length $H’$ from $[H_{\tt{min}}, H_{\tt{max}}] = [45, 55]$ and the starting positions for each segment $k_0$, $k_1$ from $[0, H-H’]$, then we cropped consecutive subsequences from the stored pair of segments as in Algorithm 1. Those hyperparameters for temporal cropping can be found in Appendix B.
>
> ---
>
> **Q2. How is the reward function learned?**
>
> **A2.**  At a high level, the reward function is learned via classifying the preferred segment by the teacher, given a pair of two segments. Specifically, we model the preference predictor $P_\psi$ in eq (1) as a softmax classifier using the accumulated sum of the reward $\psi$ over each segment for logits. By minimizing the cross-entropy loss, the learned reward $\psi$ could be consistent with the teacher’s preferences, i.e., the preferred segment gets a higher sum of reward.
> As you noted, there can be infinite equivalent reward functions with a different random seed, because the softmax function is invariant to constant offsets. It has been widely known that the RL performance is sensitive to the reward scale, however, we stabilize the training by bounding the reward function to $[−1, 1]$ using tanh function, following the previous works of preference-based RL [1].
>
> ---
>
> **Q3. Discussion on the assumption of temporal cropping.**
>
> **A3.**  We first remark that our assumption of temporal cropping naturally holds in many tasks if we don’t assume some extreme cases that break the consistent preference, e.g., applying too strong augmentation. Actually, such a cropping augmentation has been successfully utilized in computer vision tasks. It is very well-known that properly cropped images can prevent overfitting and improve the performance. We extended a similar idea to our problem setup and found that this augmentation can be useful for improving the efficiency and performance of preference-based RL for the first time.
>
>  To further address your concerns, we evaluate the case where a strong augmentation is applied and provide additional experimental results in Appendix C of the revised draft. In Walker-walk task with 100 queries, we apply the temporal cropping to PEBBLE, with a varying cropping range, i.e., $(H_{\tt{max}} - H_{\tt{min}})/2$, from an array of $[2, 5, 10, 20]$. We observe that our method consistently improves the performance of the PEBBLE. Although the performance with a large cropping range of 20 slightly underperforms the performance with 10, these results show that our method is quite robust to such noisy augmentation. In addition, we provide a discussion for another exception for assumption, sparse reward setup, in A2 for Reviewer JjVp.
>
> ---
>
> **Q4. Limited technical novelty.**
>
> **A4.**  Our contributions and novelty are stated in the introduction of the paper. They are (i) an introduction of new data augmentation, called temporal cropping, for preference-based RL and (ii) a novel combination of semi-supervised learning and the proposed data augmentation. It was not obvious before this work that data augmentations and semi-supervised learning were useful in the preference-based RL setting. In particular, we think our effective, yet simple, semi-supervised learning technique can be an important milestone in the preference-based RL literature, as an unlimited number of unlabeled data can be easily obtained with no additional cost, i.e., from past experiences stored in the buffer. We emphasized our contributions in the revised draft to make these points clearer.
>
> ---
>
> **References**
>
> [1] Kimin Lee, Laura Smith, and Pieter Abbeel. Pebble: Feedback-efficient interactive reinforcement learning via relabeling experience and unsupervised pre-training. In International Conference on Machine Learning, 2021.

---

> ### Author Response · Authors · 2021-11-22
> **A gentle reminder**
>
> Dear Reviewer qrsj,
>
> Thank you for your time and efforts in reviewing our paper.
>
> We kindly remind that the discussion period will end in 30 hours or so.
>
> We sincerely hope that our response and results of the supporting experiments successfully clarify your concerns.
>
> We just wonder whether we could have the last chance to address your further concerns or questions (if you have any).
>
> Thank you very much!
>
> Authors

---

### Author Response · Authors · 2021-11-19
**General Response**

Dear reviewers and AC,

We deeply appreciate your time and effort to review our manuscript.

Our work proposes a novel combination of semi-supervised reward learning and a new data augmentation method for feedback-efficient preference-based RL. The reviewers highlighted our strong empirical performance (qrsj, cS2w, JjVp), extensive experiments (qrsj, JjVp), and clear presentation (cS2w, JjVp), while agreeing that our topic is important (qrsj).

In response to the questions and concerns you raised, we have carefully revised and enhanced the manuscript with the following additional experiments and discussions:

- Highlighting our contributions in preference-based RL literature (Section 1)
- Additional ablation study of the semi-supervised learning and the proposed data augmentation (Figure 7 in Appendix C)
- Investigation of applying our temporal cropping augmentation with other augmentation methods (Figure 8a in Appendix C)
- Investigation of the effects of the augmentation intensity (Figure 8b in Appendix C)
- More clarification of the method description (Section 3 and 4)

These updates are temporarily highlighted in "red" for your convenience to check.

With valuable feedback from the reviewers, we sincerely believe that our work becomes even stronger and could deliver the benefits of utilizing unlabeled samples and data augmentation in preference-based RL to the audience of ICLR.

Thank you very much,

Authors.

---

### Author Response · Authors · 2021-11-19
**A gentle reminder**

Dear Reviewers,

Thank you for your time and efforts in reviewing our paper.

We kindly remind that the discussion period will end soon (in a few days). We believe that we sincerely and successfully address your concerns/questions/misunderstandings/suggestions, with the results of the supporting experiments.

If you have any further concerns or questions, please do not hesitate to let us know.

Thank you very much!

Authors

---

### Decision · Program_Chairs · 2022-01-20

**Decision:**

Accept (Poster)

**Comment:**

The topic of learning reward functions from preferences and how to do this efficiently is of high interest to the ML/RL community. All reviewers appreciate the suggested technical approach and the thorough evaluations that demonstrate clear improvements. While the technical novelty of the paper is not entirely compelling, all reviewers recommend acceptance of the paper.